# The Putative C_2_H_2_ Transcription Factor VadH Governs Development, Osmotic Stress Response, and Sterigmatocystin Production in *Aspergillus nidulans*

**DOI:** 10.3390/cells11243998

**Published:** 2022-12-10

**Authors:** Xiaoyu Li, Yanxia Zhao, Heungyun Moon, Jieyin Lim, Hee-Soo Park, Zhiqiang Liu, Jae-Hyuk Yu

**Affiliations:** 1Department of Bacteriology, University of Wisconsin-Madison, Madison, WI 53706, USA; 2School of Life Sciences, Hainan University, Haikou 570228, China; 3One Health Institute, Hainan University, Haikou 570228, China; 4Key Laboratory for Biotechnology on Medicinal Plants of Jiangsu Province, Department of Life Science, Jiangsu Normal University, Xuzhou 221116, China; 5School of Food Science and Biotechnology, Kyungpook National University, Daegu 41566, Republic of Korea; 6Department of Systems Biotechnology, Konkuk University, Seoul 05029, Republic of Korea

**Keywords:** *Aspergillus nidulans*, velvet regulators, C_2_H_2_ transcription factors, development, sterigmatocystin, stress response

## Abstract

The VosA-VelB hetero-dimeric complex plays a pivotal role in regulating development and secondary metabolism in *Aspergillus nidulans*. In this work, we characterize a new VosA/VelB-activated gene called *vadH*, which is predicted to encode a 457-amino acid length protein containing four adjacent C_2_H_2_ zinc-finger domains. Mutational inactivation of *vosA* or *velB* led to reduced mRNA levels of *vadH* throughout the lifecycle, suggesting that VosA and VelB have a positive regulatory effect on the expression of *vadH*. The deletion of *vadH* resulted in decreased asexual development (conidiation) but elevated production of sexual fruiting bodies (cleistothecia), indicating that VadH balances asexual and sexual development in *A. nidulans*. Moreover, the *vadH* deletion mutant exhibited elevated susceptibility to hyperosmotic stress compared to wild type and showed elevated production of the mycotoxin sterigmatocystin (ST). Genome-wide expression analyses employing RNA-Seq have revealed that VadH is likely involved in regulating more genes and biological pathways in the developmental stages than those in the vegetative growth stage. The *brlA*, *abaA*, and *wetA* genes of the central regulatory pathway for conidiation are downregulated significantly in the *vadH* null mutant during asexual development. VadH also participates in regulating the genes, *mat2*, *ppgA* and *lsdA*, etc., related to sexual development, and some of the genes in the ST biosynthetic gene cluster. In summary, VadH is a putative transcription factor with four C_2_H_2_ finger domains and is involved in regulating asexual/sexual development, osmotic stress response, and ST production in *A. nidulans*.

## 1. Introduction

*Aspergillus* spp. are widely distributed in nature and closely related to human life. public health, the fermentation and food processing industry including *A. oryzae* and *A. niger*, as well as some plant and human pathogens such as *A. flavus* and *A. fumigatus* [1]. The production of conidia is the primary mode of reproduction in the genus *Aspergillus*. Conidial formation is an elaborate process, which involves cell differentiation, gene expression, and signal transduction [2]. Conidia are an important carrier for the transmission and infection of *Aspergillus*, and its developmental process is also related to mycotoxin biosynthesis [3]. Therefore, elucidating the regulatory network of conidial development and exploring the functions of critical regulatory proteins are crucial for understanding the law of growth and development of *Aspergillus*, rational utilization of beneficial industrial strains and effective control of pathogens.

*Aspergillus nidulans* is an important model organism for studying fungal development, secondary metabolism, and its regulatory pathway. The mechanism of conidiation has been well studied in *A. nidulans* [4,5]. In *A. nidulans*, BrlA, AbaA, and WetA constitute the central regulatory pathway of conidial formation, which can sequentially activate conidial development and control the expression of specific genes involved in asexual development [2]. The activation of BrlA is a fundamental step to initiate conidiation [4,6]. Then, BrlA can activate AbaA, which controls the differentiation and function of phialide [7]. In the late stage, WetA is activated by AbaA, which regulates the expression of several spore-specific genes and conidial wall synthesis [8]. In addition, it is reported that velvet family proteins also play an essential role in the asexual development of *A. nidulans* [9,10].

Velvet family proteins are fungal-specific proteins with the velvet domain, including VosA, VeA, VelB and VelC, which are highly conserved in filamentous fungi [10,11]. In *Aspergillus*, velvet proteins participate in regulating growth and development, pigmentation, primary and secondary metabolism [9,12,13]. These proteins often form various complexes, such as VosA-VelB, VelB-VeA-LaeA, VosA-VelC and VelB-VelB, which play different roles in coordinating the growth, differentiation, and secondary metabolism of *Aspergillus* [12,13]. Among these complexes, the VosA-VelB complex plays a crucial role in regulating conidia maturation and germination, trehalose biosynthesis and conidial wall synthesis [13,14,15].

The VosA-VelB protein complex has a DNA-binding domain similar to that of mammalian NF-κB transcription factor, which can recognize the cis-acting motif specifically in the promoter region of its target genes [15]. Chromatin immunoprecipitation (ChIP) showed that VosA-VelB could bind to promoter sequences of more than 150 genes, including genes activated by VosA-VelB, such as *tpsA* and *treA* genes related to trehalose synthesis [15]. Another VosA/VelB-activated gene, *vadA*, is a novel bifunctional regulatory factor in *A. nidulans*, which is involved in regulating conidial germination, trehalose synthesis, β-glucan synthesis, oxidative stress, and sterigmatocystin synthesis [16]. In addition, VosA-VelB also has negative regulatory effects on many genes, including *brlA* and *wetA* in the central regulatory pathway of conidia, and it can also inhibit β-glucan synthesis in conidia and ascospores by directly binding to the promoter region of *fksA* gene encoding β-1, 3-glucan synthase [17].

While several target genes of VosA-VelB have been functionally characterized, many remain to be investigated. In this study, we have identified another VosA-VelB target gene *vadH* (VosA/VelB-Activated Developmental gene H; AN6503 AspGD). The promoter region of *vadH* is bound by both VosA and VelB in conidia. The *vadH* gene is predicted to encode a highly conserved transcription factor (VadH) harboring four C_2_H_2_ zinc finger domains. The homologous proteins of VadH are universal in the fungal kingdom, and some have been well characterized. *Saccharomyces cerevisiae* Azf1, a homolog of VadH, participates in activating the genes related to carbon and energy metabolism when glucose exists, and switches to maintaining cell wall integrity when glucose is depleted [18]. In *Magnaporthe oryzae*, Cos1 is involved in regulating the development of conidiophores and melanin biosynthesis [19]. The VadH homolog CgAzf1 coordinates melanin production, conidium development, appressorium formation and virulence in *Colletotrichum gloeosporioides* [20]. In this work, the biological functions of *vadH* have been characterized by gene knockout, overexpression, and transcriptome analyses.

## 2. Materials and Methods

### 2.1. Strains and Culture Conditions

All fungal strains used in this study are listed in Appendix A [13,21,22], and media are prepared as previously described [9]. Briefly, minimal media with glucose (MMG) and MMG with 0.5% yeast extract (MMYE) were used for general purposes, and sexual medium (SM) was used for enhancing sexual development. To determine the number of conidia and cleistothecia, the wild type (WT: *A. nidulans* FGSC4), mutants, and complemented strains were inoculated and cultured on solid MMG, MMYE or SM for seven days at 37 °C. Micrographs were taken by a Zeiss M2Bio microscope. For the overexpression of *vadH* under the *niiA* promoter, strains were inoculated on a non-inducing medium (MMG containing 0.2% ammonium tartrate as a nitrogen source) or an inducing medium (MMG containing 0.6% sodium nitrate as a nitrogen source) and incubated at 37 °C for three days.

### 2.2. Nucleic Acid Isolation and Manipulation

Total RNA isolation was performed as previously described [23]. For asexual and sexual development, cultures from mutants and WT were collected and transferred on solid MMG and SM, respectively. Then, the plates were exposed to air for asexual developmental induction or tightly sealed and blocked from light for sexual developmental induction. For asexual development, samples were collected at 18 and 36 h for RNA isolation. Samples were harvested at 36 and 72 h after transfer for sexual development. For vegetative growth, one milliliter of conidial suspension (10^6^ conidia/mL) was inoculated in 100 mL liquid MMG and incubated at 37 °C. The mycelium was collected at 18 and 36 h post-inoculation (hpi) for RNA isolation. Quantitative reverse transcription-PCR was used to analyze the expression levels of *vadH*. The primers were listed in Appendix A. The quantitative reverse transcription PCR (qRT-PCR) was carried out by the Fast SYBR Green Master Mix. Gene expression levels were normalized using the endogenous control gene actin. The average normalized expression level was calculated using the 2^−ΔΔCt^ method [24]. All the experiments were repeated three times.

### 2.3. Target Deletion of VadH

Genomic DNA extraction was performed as previously described [25,26]. The primers used in this section are listed in Appendix A. The *vadH*-deletion mutant strain (Δ*vadH*) was generated by double-joint PCR (DJ-PCR) as previously described [23]. The up- and down-stream sequences of the *vadH* gene were amplified from *A. nidulans* FGSC4 genomic DNA the by PCR using the primer pairs OXL-1/OXL-2 and OXL-3/OXL-4. The *A. fumigatus pyrG* marker was amplified using the primer pair OHS-694/OHS-695 from *A. fumigatus* AF293 genomic DNA. The *vadH* deletion cassette was amplified with primer pair OXL-5/OXL-6 and introduced into *A. nidulans* RJMP1.59 [21]. Protoplasts were generated from *A. nidulans* RJMP1.59 by the Vinoflow FCE lysing enzyme (Novozymes) [25]. For the complementation of Δ*vadH*, the *vadH* gene sequence, including its predicted promoter, was amplified with the primer pair OXL-15/OXL-16 and attached to a pHS13 vector [13]. To generate the overexpressing strain, the *vadH* ORF derived from the genomic DNA was amplified using the primer pair OXL-31/OXL-32. The PCR product was then attached into pHS11 and introduced into *A. nidulans* RJMP1.59. The *vadH*-overexpressing strains were screened by qRT-PCR.

### 2.4. Spore Viability Determination

To test spore viability, conidia obtained from two-day-old cultured WT, mutant and complementary strains were spread on solid MMG and cultured at 37 °C. Then, conidia were collected after culturing for seven days. About 100 conidia were coated onto solid MMG and incubated at 37 °C for 48 h in triplicate. Survival rates were determined as the ratio of the number of viable colonies to the number of conidia inoculated.

### 2.5. Osmotic Stress Assays

For stress tests, the strains were inoculated on the solid MMG medium including sorbitol (1.0 M), glycerol (1.0 M) and NaCl (1.0 M) and grown at 37 °C for seven days. Moreover, ten microliters of serially diluted conidia suspensions (10 to 10^5^ conidia/mL) were spotted on the solid MMG medium including sorbitol (1.0 M), glycerol (1.0 M) and NaCl (1.0 M), and incubated at 37 °C for 48 h. The plates without the stress factors served as controls. The inhibition rates were calculated as follows: Inhibition rate (%) = (D_ck_ − D_t_)/D_ck_ × 100%. Where D_ck_ is the colony diameter of the strain in the control, and D_t_ is the colony diameter of the strain in the treatment group.

### 2.6. Sterigmatocystin (ST) Determination

ST extraction and determination were performed as previously described [16]. Briefly, 10^6^ conidia from the strains were inoculated in 2 mL liquid MMG and cultured at 37 °C for seven days. ST was extracted by adding 2 mL of CHCl_3_. The organic phase (CHCl_3_) was separated through centrifugation and transferred to a new glass bottle. The extracting solution was evaporated in a fume hood and dissolved in 1 mL acetonitrile:methanol (50:50, *v*/*v*). After filtering through a millipore filter (0.45 μm), the samples were analyzed by high-performance liquid chromatography with diode-array detection (HPLC-DAD, Agilent Technologies, Waldbronn, Germany).

### 2.7. RNA Sequencing (RNA-Seq)

The isolation of RNA samples was performed as previously described [23]. RNA-Seq analyses of VadH include three aspects: vegetative growth, asexual and sexual development. The preparation of samples in three stages was conducted as previously described [9]. Samples for vegetative growth were collected at 18 hpi, and samples for asexual and sexual development were obtained at 18 and 36 hpi, respectively. A MGISEQ-2000 platform (BGI, Shenzhen, China) was used to analyze the samples. The genome of *A. nidulans* FGSC A4 from AspGD (http://www.aspergillusgenome.org/, accessed on 1 Septemper 2020) was used as a reference. Data processing and analyses were performed as described previously [27]. The results of RNA-Seq were verified by qRT-PCR according to the published method [27].

### 2.8. Statistical Analysis

Statistical differences between WT and mutant strains were evaluated by Student’s unpaired *t*-test. Data are reported as mean ± SD, and statistical significance was defined as *p* < 0.05.

## 3. Results

### 3.1. Characterization of VadH

The gene *vadH* (AN6503) is predicted to encode a 457-amino acids (aa) protein, which contains four adjacent C_2_H_2_ zinc-finger domains from the position 239 to 352 (Figure 1A). The homologous proteins of VadH are ubiquitous in fungi, and all of them harbor four C_2_H_2_-type domains (Figure 1A,B). VadH is homologous with CgAzf1 in *Colletotrichum gloeosporioides* [20], Cos1 in *Magnaporthe oryzae* [19], and Azf1 in *Saccharomyces cerevisiae* [18].

To evaluate the effects of VosA and VelB on the expression of *vadH*, mRNA levels of *vadH* were determined in the Δ*vosA* and Δ*velB* strains. As shown in Figure 2A, the expression levels of *vadH* in Δ*vosA* and Δ*velB* are significantly lower than those of WT in different stages, suggesting that VosA and VelB can positively regulate the expression of *vadH*. In the wild type, the expression levels of *vadH* during asexual and sexual development are apparently higher than those in the stage of vegetative growth (Figure 2B).

### 3.2. VadH Balances Asexual and Sexual Development

To study the function of *vadH*, we generated the *vadH*-deletion mutant (Δ*vadH*) and the complemented strain (C’*vadH*), and the results related to the verification of the mutant were shown in Appendix A. For colony growth, no significant difference was found between Δ*vadH* and the wild type on MMG and MMYE, whereas the colony color of Δ*vadH* was much lighter than that of strain WT (Figure 3A,B). Δ*vadH* produced fewer conidia than WT on both MMG and MMYE (Figure 3C), and the spore viability of Δ*vadH* was also slightly lower than that of the WT (Figure 3D). Compared with WT, the mutant Δ*vadH* exhibited significantly decreased conidial germination (Appendix A). For sexual development, WT, Δ*vadH* and C’*vadH* were inoculated on SM, and the number of cleistothecia was determined (Figure 3E). As shown in Figure 3F, the mutant Δ*vadH* produced significantly more cleistothecia than WT and Δ*vadH* on MMG, and MMYE, suggesting that VadH is involved in regulating sexual development in *A. nidulans*. However, there was no significant difference in cleistothecia yields on SM between WT and Δ*vadH*.

### 3.3. Disruption of VadH Led to Elevated Sensitivity to Osmotic Stress

For osmotic stress assays, the strains were inoculated on the solid MMG including sorbitol (1.0 M), glycerol (1.0 M) and NaCl (1.0 M), and cultured at 37 °C for 7 days. As shown in Figure 4A,B, Δ*vadH* was more sensitive to hyper-osmotic stress than WT, and it had an obvious difference in colony color on the stress plates. Δ*vadH* hardly formed conidia on the MMG with 1.0 M NaCl and produced noticeably fewer conidia than the wild type on the MMG with 1.0 M glycerol and 1.0 M sorbitol. Regarding conidial suspension assays, 10^2^ conidia of Δ*vadH* could not form a colony on the MMG plates with glycerol (1.0 M), sorbitol (1.0 M) and NaCl (1.0 M) (Figure 4C). These results propose that the *vadH* gene is involved in regulating osmotic stress response of *A. nidulans*.

### 3.4. Deleting VadH Leads to an Increase in ST Production

To test the effect of *vadH* on ST production, ST was extracted from WT, Δ*vadH* and C’*vadH* and analyzed using HPLC. As shown in Figure 5, the Δ*vadH* mutant produced more ST than WT and C’*vadH*, suggesting that VadH may negatively regulate ST production in *A. nidulans*.

### 3.5. Overexpression of VadH Suppresses Sexual Development

As mentioned above, the deletion of *vadH* leads to enhanced sexual development on MMG and MMYE. To further validate the function of VadH in fungal development, we generated the *vadH*-overexpression strain (OE*vadH*) and analyzed its phenotypes (Figure 6A). On the non-inducing medium, there was no significant differences in the number of conidia and cleistothecia between OE*vadH* and WT (Figure 6B,C). However, when induced, overexpression of *vadH* led to significantly reduced production of cleistothecia (Figure 6C). The expression level of vadH in OE*vadH* on the inducing medium was verified by qRT-PCR (Appendix A). Collectively, these results suggest that VadH is essential for sexual development, and it may act as a suppressor of sexual development.

### 3.6. Transcriptomic Analyses of VadH

Genome-wide expression analyses of WT and Δ*vadH* under three stages were performed by employing RNA-Seq. The related data have been submitted to GenBank (PRJNA905844). For the vegetative growth stage, 895 DEGs were obtained, in which 328 DEGs were upregulated and 567 DEGs were downregulated. Nevertheless, for asexual and sexual development stages, more than 1200 DEGs were identified, suggesting that VadH is involved in regulating more genes in the developmental stages than those in vegetative growth (Figure 7A). The Venn diagram indicates that 126 DEGs, including 63 up-regulated and 63 down-regulated DEGs, participate in the vegetative growth, asexual and sexual development stage simultaneously (Figure 7B). The RNA-Seq results in the three stages were verified by qRT-PCR, and the expression levels of selected DEGs from three stages showed the same trend as those in RNA-Seq with all the correlation coefficients being more than 95% (Appendix A).

Based on the RNA-Seq results, we further analyzed the expression of DEGs related to sporulation and ST biosynthesis. As shown in Figure 7C, *brlA*, *abaA* and *wetA* in the central regulatory pathway are downregulated significantly in the asexual development stage of Δ*vadH*, especially *abaA*. In the process of sexual development, the genes *mat2* and *ppgA* related to sexual sporulation are upregulated, and *lsdA*, *vadJ* and *vadZ* are downregulated in Δ*vadH*; some of the genes (*stuA*, *stuL*, *stuO*, *stuQ*, *stuS*, *stuU*) in the gene cluster of ST biosynthesis are also upregulated to varying degrees.

Then, we performed the KEGG pathway analysis for DEGs in three stages (Figure 8). In the vegetative growth stage, six significant enrichment pathways were obtained, which were mainly related to amino acid metabolism and aflatoxin biosynthesis (Figure 8A). Regarding asexual and sexual development, more biological pathways were affected by VadH, including some pathways related to the metabolism of fatty acid, nitrogen and carbohydrate (Figure 8B,C). The numbers of significant enrichment pathways in the asexual and sexual development are more than twice as much as those in the vegetative growth stage.

## 4. Discussion

The velvet proteins are fungal-specific transcript factors that coordinate both development and secondary metabolism [11,15]. Previous studies indicated that the target genes regulated by VosA/VelB complex could be divided into VosA/VelB-activated developmental genes (VADs) and VosA/VelB-inhibited developmental genes (VIDs), and many VADs and VIDs are regulatory factors involved in the development process of *A. nidulans* [15,17]. Here, we characterized a new VAD gene *vadH* in *A. nidulans*, which encodes a zinc finger protein with four adjacent C_2_H_2_-type domains. The expression levels of *vadH* were noticeably decreased in the mutants Δ*vosA* or Δ*velB*, suggesting that both VosA and VelB have a positive regulatory effect on *vadH*. It is observed that the homologous proteins of VadH can play various roles in different fungi [18,19,20]. In *A. nidulans*, VadH is involved in asexual and sexual development, osmotic stress response and ST production.

Our study indicated that VadH could balance the asexual and sexual development of *A. nidulans*. VadH could inhibit conidial production and stimulate sexual development on MMY and MMYE, and overexpression of *vadH* obviously suppresses the formation of cleistothecia. VadH can negatively regulate the expression of two genes related to sexual development, *mat2* and *ppgA*, which may affect cleistothecium formation to a certain extent. It has been reported that Mat2 and PpgA can act as activators in the sexual development of *A. nidulans* [28]. In addition, VadH can also positively regulate the expression of *lsdA*, *vadJ* and *vadZ*, which have proved to function as repressors in sexual sporulation [29,30,31]. In *A. nidulans*, the central regulatory pathway activates conidial development and regulates the expression of specific genes in conidiation. In the mutant Δ*vadH*, the genes *brlA*, *abaA*, and *wetA* are all downregulated in the asexual development stage, which will inhibit the conidial production of Δ*vadH*. Especially *abaA* related to the differentiation of phialide is downregulated dramatically, which may affect the formation of conidia to a great extent. Therefore, VadH may function as a positive regulator in asexual sporulation and a negative one in sexual development. It is reported that VadA (AN5709), a member of VADs, also participates in the balance between asexual and sexual development [16]. The *ΔvadA* mutant exhibits increased production of cleistothecia, and overexpression of *vadA* leads to increased conidial production. Similar to *vadH*, the other two VADs, VadJ (AN3214, a histidine kinase) and VadZ (AN8774, a C_6_ transcription factor) also act as activators of asexual development and repressors of sexual development [30,31]. Interestingly, one member of VIDs, VidA (AN2498) proved to be essential for proper asexual and sexual development in *A. nidulans*. Deletion of *vidA* can lower the production of conidia, and slightly increases the yield of cleistothecia [32]. These reported genes regulated directly by the VosA-VelB complex all have a function of balancing asexual and sexual development, suggesting that the complex does play a crucial role in the development process of *A. nidulans*.

VadH participates in the osmotic pressure response of *A. nidulans*. Deletion of *vadH* leads to elevated sensitivity to hyperosmotic stress, and the asexual sporulation of Δ*vadH* is also suppressed significantly on the hyperosmotic media. The KEGG analysis shows that the expression of more than 60 genes in the mitogen-activated protein kinase (MAPK) signaling pathway is influenced by VadH, and many of them are involved in cell wall integrity and high osmolarity pathways. We speculate that deleting vadH may change cell wall integrity, subsequently affecting the osmotic pressure response of *A. nidulans*. In *S. cerevisiae*, Azf1 also have the function of maintaining cell wall integrity when glucose is depleted [18]. However, the VadH orthologues, Cos1 and CgAzf1 are not related to osmotic pressure response in *M. oryzae* and *C. gloeosporioides*, respectively [19,20]. Further study will be needed to elucidate the precise mechanism of VadH coordinating the hyperosmotic stress. Regarding secondary metabolism, VadH proves to be involved in regulating ST production. Disruption of *vadH* leads to elevated ST production, and the RNA-Seq results also indicate that VadH can negatively regulate the expression of several genes related to the biosynthesis of ST. It has been found that Cos1 and CgAzf1 are also relevant to secondary metabolism, and both of them can regulate melanin production [19,20]. Similar to *vadH*, the VADs, *vadA*, *vadJ* and *vadZ*, also exhibit the function of suppressing ST production [16,30,31], whereas the VID gene *vidA* is not involved in the biosynthesis of ST [32].

## 5. Conclusions

Taken together, we propose a working model for VadH regulating asexual/sexual development and secondary metabolism (Figure 9). The VosA-VelB complex can positively regulate the expression of *vadH*. Then, VadH functions as a positive regulator for asexual sporulation through the central regulatory pathway, and it also acts as a negative regulator for the production of cleistothecia and sterigmatocystin. In summary, a newly identified VAD gene *vadH* is predicted to encode a C_2_H_2_-type transcription factor, which is involved in balancing asexual/sexual development and regulating osmotic stress and sterigmatocystin production. These findings provide further evidence for the crucial roles of VADs in the development and secondary metabolism of *A. nidulans*. Understanding the mechanism of VadH will contribute to revealing the precise regulatory networks of the VosA-VelB complex.

## Figures and Tables

**Figure 1 cells-11-03998-f001:**
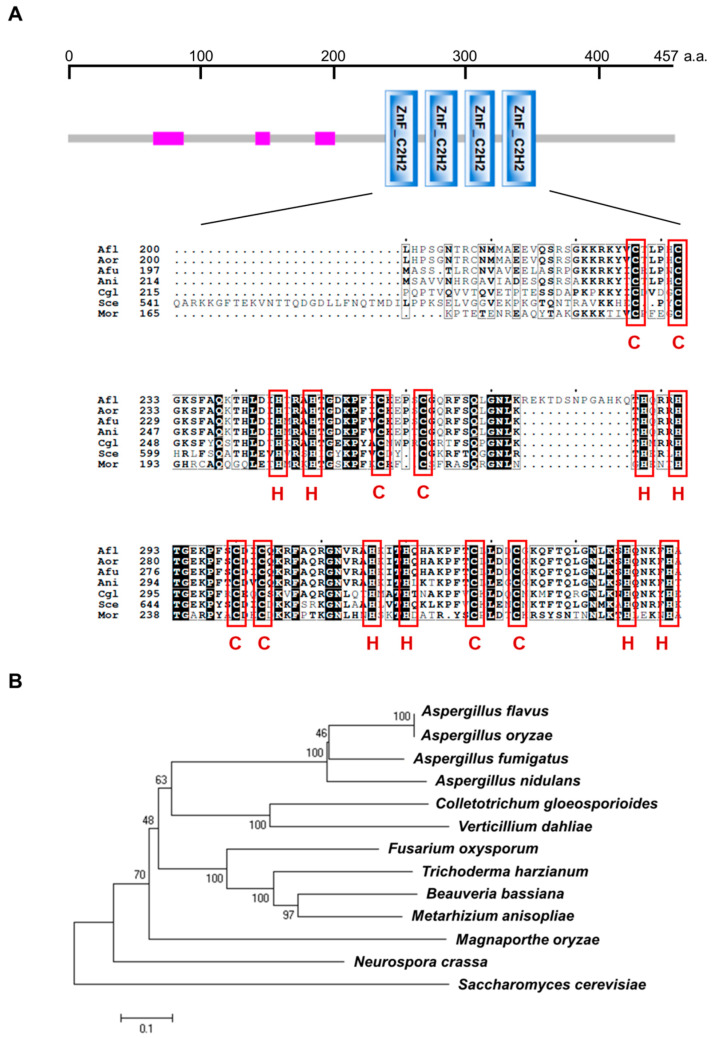
Overview of the VadH protein. (**A**) Protein domain analysis of VadH. The pink rectangle is a region of low compositional complexity. The below figure presents the alignment of C_2_H_2_ zinc-finger domains from *A*. *nidulans* VadH (Ani: AN6503) and its homologs from other fungi. (**B**) Phylogenetic analysis of VadH. The phylogenetic tree was constructed using Clustal_W and MEGA 6.0 with homologous sequences of VadH from *A. flavus* (Afl: AFL2T_05677), *A. oryzae* (Aor: AO090701000019), *A. fumigatus* (Afu: Afu6g05160), *Colletotrichum gloeosporioides* (Cgl: AUS82351.1), *Verticillium dahlia* (XP_009658472.1), *Fusarium oxysporum* (EWZ45422.1), *Trichoderma harzianum* (KKP06506.1), *Beauveria bassiana* (XP_008598185.1), *Metarhizium anisopliae* (KFG78752.1), *Magnaporthe oryzae* (Mor: XP_003719876.1), *Neurospora crassa* (XP_961139.2) and *Saccharomyces cerevisiae* (Sce: NP_014756.3).

**Figure 2 cells-11-03998-f002:**
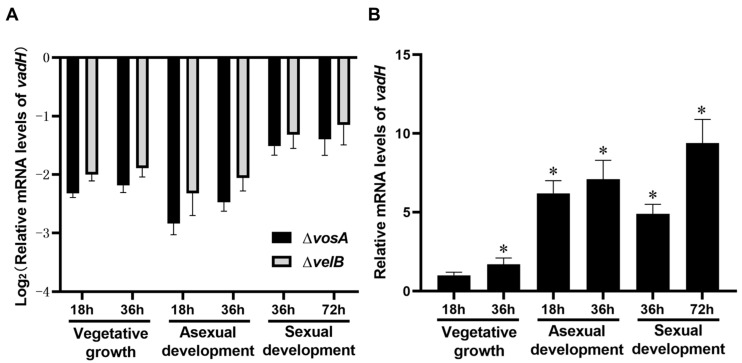
Relative mRNA levels of *vadH*. (**A**) Relative expression levels of *vadH* in Δ*vosA* and Δ*velB*. (**B**) Relative expression levels of *vadH* in WT during three stages. The expression level of *vadH* at 18 h of the vegetative growth stage was set to 1. The asterisks represent significant level (* *p* < 0.05).

**Figure 3 cells-11-03998-f003:**
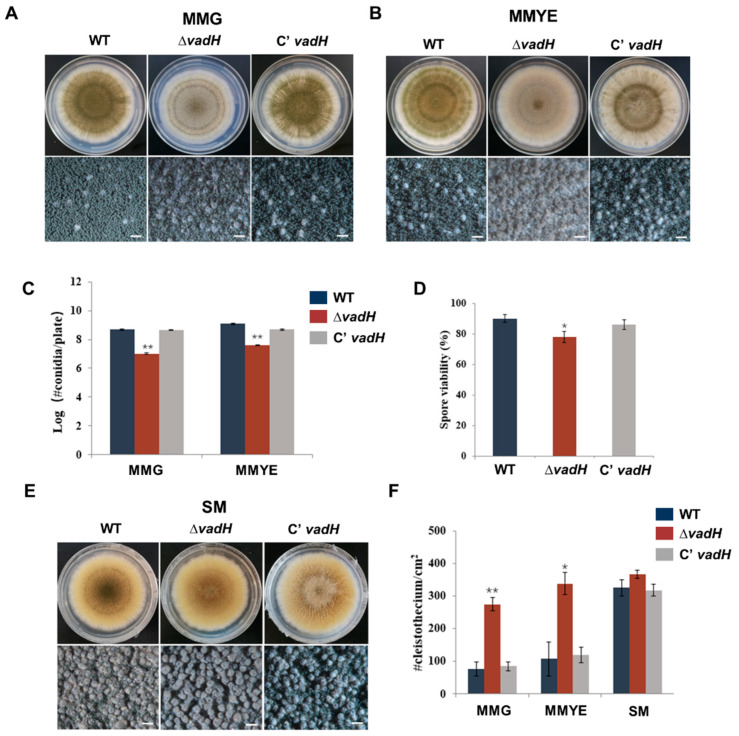
Developmental phenotypes of the Δ*vadH* mutant. (**A**,**B**) Colony morphology of WT, Δ*vadH*, and the complemented strain on MMG (**A**) or MMYE (**B**) and grown for 7 days. The bottom panel shows close-up views of the colony in the top panel (Bar = 0.5 mm). (**C**) Statistical analyses of conidial yields on MMG and MMYE. (**D**) Statistical analyses of spore viabilities. (**E**) Colony morphology of WT, Δ*vadH*, and the complemented strain on SM and grown for 7 days. The bottom panel shows close-up views of the colony in the top panel (Bar = 0.5 mm). (**F**) Statistical analyses of cleistothecia on MMG, MMYE and SM. The asterisks represent significant level (* *p* < 0.05, ** *p* < 0.001).

**Figure 4 cells-11-03998-f004:**
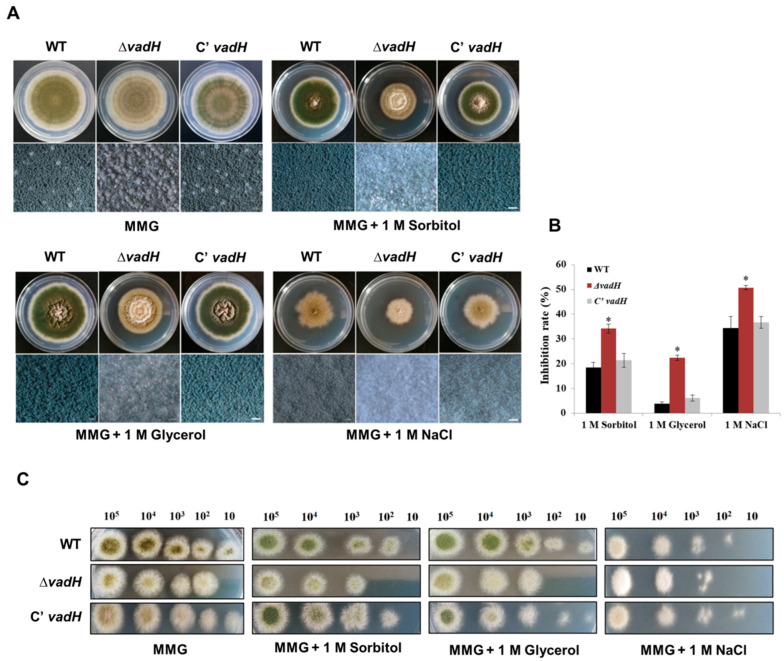
Effects of the hyperosmotic medium on the growth of WT, Δ*vadH* and the complemented strain. (**A**) Comparison of the growth of strains on the MMG added with 1.0 M Sorbitol, 1.0 M Glycerol and 1.0 M NaCl and cultured at 37 °C for 7 days. The bottom panel shows close-up views of the colony in the top panel (Bar = 0.5 mm). (**B**) The inhibition rate of stress factors against WT, Δ*vadH* and the complemented strain. The asterisks represent significant level (* *p* < 0.05). (**C**) The growth of serially diluted conidia on the MMG added with 1.0 M Sorbitol, 1.0 M Glycerol and 1.0 M NaCl, culturing at 37 °C for 2 days.

**Figure 5 cells-11-03998-f005:**
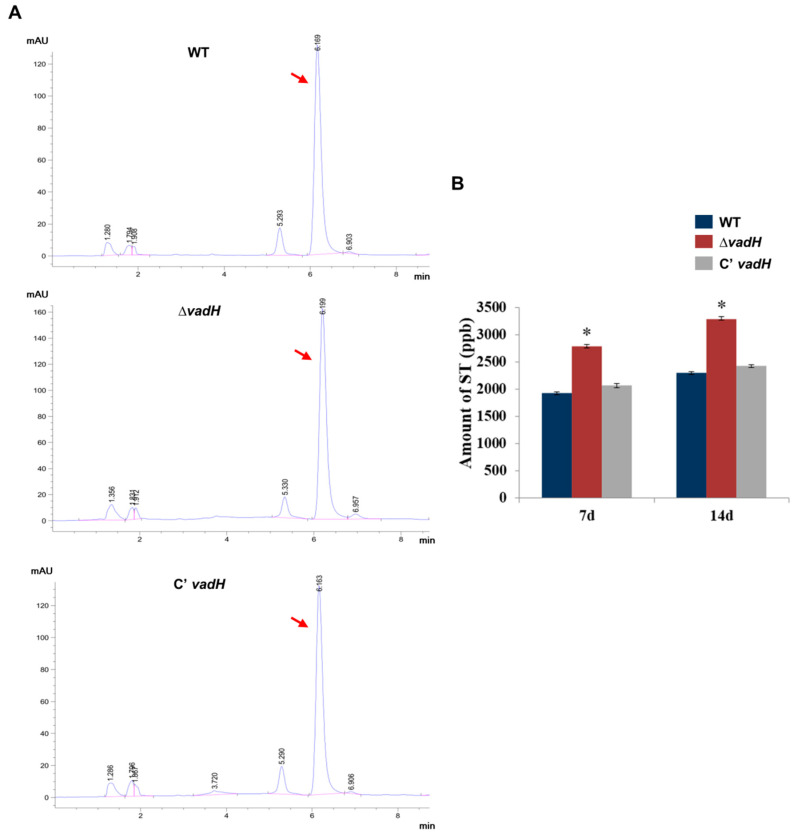
HPLC analysis of ST yield. (**A**) Determination of ST production of WT, Δ*vadH* and the complemented strain using HPLC. Arrows indicate the peak of ST. (**B**) Statistical analyses of ST yields. The asterisk represents significant level (* *p* < 0.05).

**Figure 6 cells-11-03998-f006:**
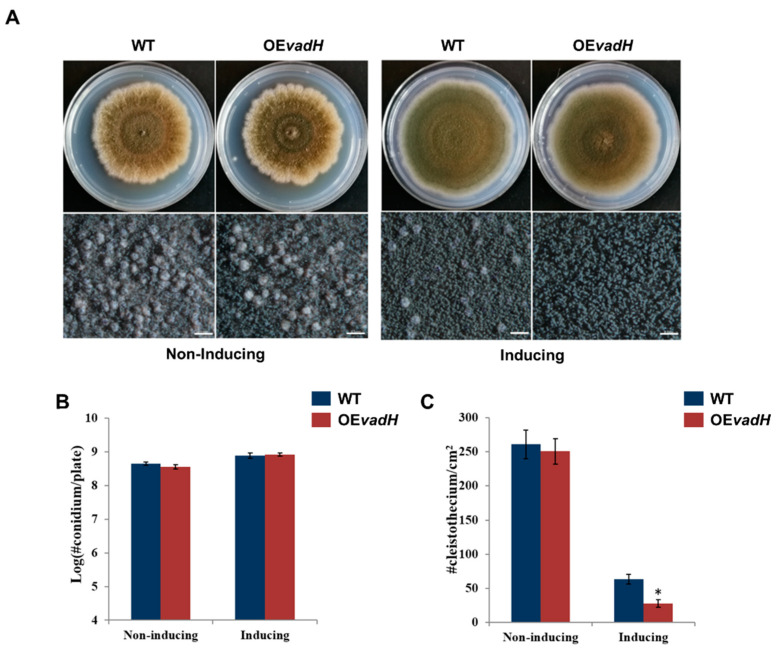
Effects of overexpression of *vadH* on the development of *A. nidulans*. (**A**) WT and OE*vadH* were inoculated onto solid MMG (non-inducing) and MMT (100 mM threonine, inducing) and cultured at 37 °C for 7 days. The bottom panel shows close-up views of the colony in the top panel (Bar = 0.5 mm). (**B**) Statistical analyses of conidial yields under non-induced and induced conditions. (**C**) Statistical analyses of cleistothecia under non-induced and induced conditions. The asterisk represents significant level (* *p* < 0.05).

**Figure 7 cells-11-03998-f007:**
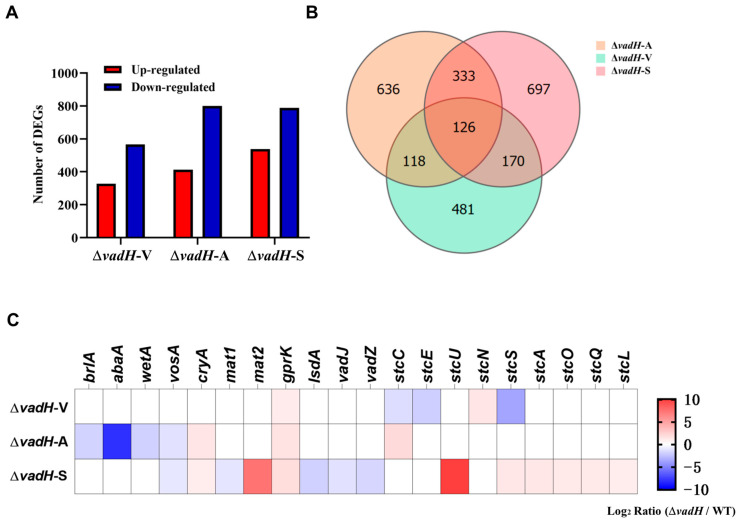
Results of RNA-Seq analyses. (**A**) DEGs statistics of Δ*vadH* at in three stages. Δ*vadH*-V: vegetative growth, Δ*vadH*-A: asexual development, Δ*vadH*-S: sexual development. (**B**) A Venn diagram of DEGs in the three stages. (**C**) A heatmap of partial DEGs involved in asexual/sexual development and ST production.

**Figure 8 cells-11-03998-f008:**
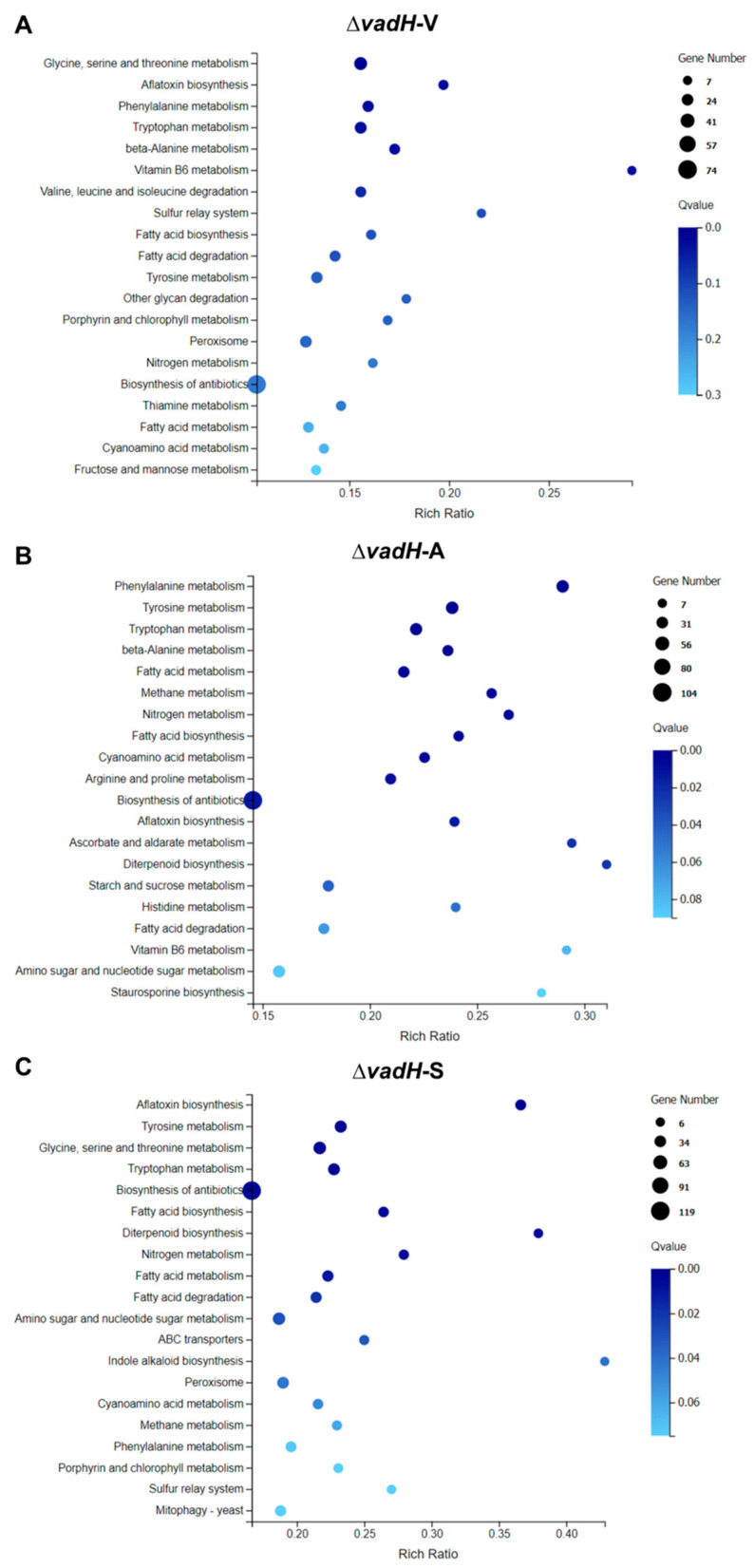
KEGG pathway enrichment of DEGs during the vegetative growth (**A**), asexual developmental (**B**), and sexual developmental (**C**) phases.

**Figure 9 cells-11-03998-f009:**
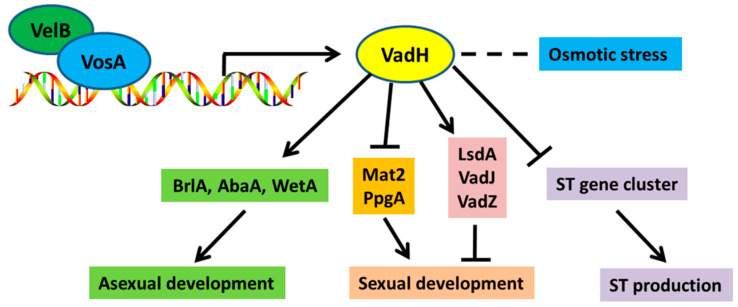
A genetic model of VadH regulating asexual/sexual development and secondary metabolism of *A. nidulans*.

## Data Availability

The data presented in this study are available on request from the corresponding author.

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
