# Peer review of "The Putative C2H2 Transcription Factor VadH Governs Development, Osmotic Stress Response, and Sterigmatocystin Production in Aspergillus nidulans"

_cells, 2022, doi:10.3390/cells11243998_

Round 1

Reviewer 1 Report

The manuscript of Li et al. described the role of C2H2 transcription factor VadH in A. nidulans, and found VadH is involved in fungal asexual and sexual development, osmotic stress response, and sterigmatocystin production. The result is meaningful and should be published. However, the manuscript could be improved somewhere before publication.

Figure 2A, Use logarithmic coordinates for the ordinate (Relative expression levels of vadH).

Lines 215-224, Where is the evidence for disruption of vaDH?

Figure 3A,E, What are 3A and 3E's lower panels? Please to indicate clearly.

Figure 4 A, What are 4A's lower panels? Please to indicate clearly. Please calculate relative growth rates of strains in different hyperosmotic media, compared to MMG.

Lines 240-248, Has the overexpression of vadH been confirmed at the mRNA or protein level?

Figure 7C, What do the values represent? Unit? Or compared with what?

Figure 8, The text in the pictures is not clear enough.

Author Response

Comments and Suggestions for Authors

The manuscript of Li et al. described the role of C2H2 transcription factor VadH in A. nidulans, and found VadH is involved in fungal asexual and sexual development, osmotic stress response, and sterigmatocystin production. The result is meaningful and should be published. However, the manuscript could be improved somewhere before publication.

Figure 2A, Use logarithmic coordinates for the ordinate (Relative expression levels of vadH).

Figure 2A has been revised by using logarithmic coordinates.

Lines 215-224, Where is the evidence for disruption of vaDH?

The evidence for disruption of vadH has been displayed in supplementary material Figure S1.

Figure 3A,E, What are 3A and 3E's lower panels? Please to indicate clearly.

Figure 3A and 3E's lower panels indicate close-up views of the colony in top panel, and we have clarified it in Figure 3.

Figure 4 A, What are 4A's lower panels? Please to indicate clearly. Please calculate relative growth rates of strains in different hyperosmotic media, compared to MMG.

We have clarified the lower panels in Figure 4. A figure about inhibition rates has been added in Figure 4B.

Lines 240-248, Has the overexpression of vadH been confirmed at the mRNA or protein level?

The overexpression of vadH has been verified by qRT-PCR, and the result has been displayed in supplementary material Figure S3.

Figure 7C, What do the values represent? Unit? Or compared with what?

The values represent relative expression levels [log2 Ratio (DvadH/WT)] in the RNA-Seq data, and it has been added in Figure 7C.

Figure 8, The text in the pictures is not clear enough.

The clear pictures have been added in Figure 8.

Reviewer 2 Report

In this manuscript, the author identified a VosA/VelB-activated gene, vadH, involved in the regulation of asexual/sexual development, osmotic stress response and the mycotoxin sterigmatocystin production in Aspergillus nidulans. RNA-seq analysis showed that the deletion of vadH mutant significantly altered the expression of  genes involved in related biological processes compared to that of the WT strain. This is an interesting study and data can support conclusions.Writing is mostly clear and logic. However, Ms could be improved and some revised suggestions are followed:

1. There should have the previous published information for other fungal species, such as Saccharomyces cerevisiae, etcfor functions of vadH in the introduction section. 

2. In Fig. 1B phylogenetic tree, there was no information about S. cerevisiae Please check phylogenetic tree with legend and keep it consistent.

3.Significant differences (p-value) in Fig. 2 should be indicated.

4.In Fig. 3, does spore viability correlate with germination? If there have microscopic images for germlings, it should be a better way to be displayed. 

5. In Fig. 6, what is the expression level of OE::vadH under the induction conditions. 

6.Please labeling description of  values in Fig. 7C.

7.Figure 8 image quality needs to be improved.

8.The font size is inconsistent in whole manuscript.

9.RNA-seq raw data and analysis data sets should be uploaded. 

Author Response

Comments and Suggestions for Authors

In this manuscript, the author identified a VosA/VelB-activated gene, vadH, involved in the regulation of asexual/sexual development, osmotic stress response and the mycotoxin sterigmatocystin production in Aspergillus nidulans. RNA-seq analysis showed that the deletion of vadH mutant significantly altered the expression of  genes involved in related biological processes compared to that of the WT strain. This is an interesting study and data can support conclusions.Writing is mostly clear and logic. However, Ms could be improved and some revised suggestions are followed:

  1. There should have the previous published information for other fungal species, such as Saccharomyces cerevisiae, etc. for functions of vadH in the introduction section.

The previous published information for vadH orthologs in other fungal species has been added in the introduction section.

  1. In Fig. 1B phylogenetic tree, there was no information about S. cerevisiae Please check phylogenetic tree with legend and keep it consistent. Saccharomyces cerevisiae cerevisiae have been added in the phylogenetic tree.
  2. Significant differences (p-value) in Fig. 2 should be indicated.

Significant differences have been supplied in Fig. 2.

  1. In Fig. 3, does spore viability correlate with germination? If there have microscopic images for germlings, it should be a better way to be displayed.

Yes, the spore viability correlate with germination in the mutant. DvadH exhibited decreased conilial germination. The related results have been added in supplementary material Figure S2.

  1. In Fig. 6, what is the expression level of OE::vadH under the induction conditions.

The related result has been added in supplementary material Figure S3.

  1. Please labeling description of values in Fig. 7C.

It has been added in Fig. 7C.

  1. Figure 8 image quality needs to be improved.

The clear pictures have been added in Figure 8.

  1. The font size is inconsistent in whole manuscript.

We have unified the font size in whole manuscript.

  1. RNA-seq raw data and analysis data sets should be uploaded.

The data have been uploaded in GenBank.

Reviewer 3 Report

I have reviewed the manuscript titled “The putative C2H2 transcription factor VadH governs the development, osmotic stress response, and sterigmatocystin production in Aspergillus nidulans. The work is interesting and written well.

I am giving a few suggestions to improve the texture of the manuscript.

There are many typo errors in the manuscript that should be thoroughly checked e.g., L119 A. fumigates

Alignment and font are not uniform which should be corrected. E.g., L326.

Figure 8 is not in good clarity, please change it.

Author Response

I have reviewed the manuscript titled “The putative C2H2 transcription factor VadH governs the development, osmotic stress response, and sterigmatocystin production in Aspergillus nidulans. The work is interesting and written well.

I am giving a few suggestions to improve the texture of the manuscript.

There are many typo errors in the manuscript that should be thoroughly checked e.g., L119 A. fumigates

The typo errors have been revised in the manuscript.

Alignment and font are not uniform which should be corrected. E.g., L326.

The font has been unified in Figure 1A.

Figure 8 is not in good clarity, please change it.

We have changed the pictures of Figure 8.